# Barriers to the uptake of eye health services of the children in rural Bangladesh: A community-based cross-sectional survey

A. H. M. Enayet Hussain[1]*, Labida Islam[2,3], Saidur Rahman Mashreky[2,3], A. K. M. Fazlur Rahman[2,3], Eija Viitasara[1], Koustuv Dalal[1]

**1** Department of Health Sciences, Division of Public Health Science, Mid Sweden University, Sundsvall, Sweden, **2** Centre for Injury Prevention and Research Bangladesh, Dhaka, Bangladesh, **3** Bangladesh University of Health Sciences, Dhaka, Bangladesh

* enayet.hussain@miun.se

**Data Availability Statement:** All relevant data are within the manuscript and its Supporting Information files.

## Abstract

Globally, ocular morbidity and disability among children are major public health concerns. This study was designed to explore the health-seeking behaviours of parents in Bangladesh whose children have ocular problems. A cross-sectional mixed method was followed for this study. The method was designed to measure the eye health care-seeking practices of care-givers/parents with children with ocular morbidity in three unions (the lowest administrative geographical area comprising 30,000–50,000 population) of the Raiganj Upazila under the Sirajganj District of Bangladesh. The study period was from January to April 2017. Face-to-face interviews using a semi-structured quantitative questionnaire with the caregivers and KI were conducted among the health service providers during the study period. This was the first community-based study conducted in Bangladesh to find out caregivers' health-seeking behaviour with identified ocular morbidity. Among 198 confirmed cases of childhood ocular problems, only 87 (43.9%) parents sought health care for their children's ocular morbidities. Better health-seeking behavior was found among the wealthier families. Proportions were 55.3% and 36% among wealthy and low-income families, respectively. Affluent families sought care from qualified service providers. Educated household heads chose qualified service providers for their children at a higher rate than illiterate household heads. Lack of knowledge, lack of awareness and financial constraints are significant barriers to seeking proper health care. More than half of the caregivers did not seek any eye care services for their children. Socio-demographic factors, and financial constraints play an essential role in the health-seeking behaviour of the parents.

## Introduction

Globally, ocular morbidity and disability among children are major public health concerns. Infancy and childhood are critical periods for visual development. When problems arise, children may be unable to complain or convey their difficulties, and caregivers may not appreciate

**Funding:** The author(s) received no specific funding for this work.

**Competing interests:** The authors have declared that no competing interests exist.

or understand the potential for children's visual problems. Delays in interventions may lead to mild to severe permanent visual impairments.

Globally, around 1.4 million children aged 0–14 years live with blindness, while about 17.5 million are at risk for low vision [1]. However, most of these children with visual impairments live in low and middle-income countries with a prevalence of 1.5 per 1000 children, compared with 0.3 per 1000 in high-income countries [2]. Recent studies in India indicated that more than 30% of the blind population lost their vision before the age of 17 years, and, unexpectedly, many of them lost their sight when they were less than five years [3].

In low and middle-income countries, optimal eye care is not always available and there are excessively high numbers of cases of avoidable blindness. Globally, 90% of people with blindness live in LMIC, with the most affected disadvantaged and vulnerable communities. This reflects the link between eye health problems, poverty, and low education [4]. Proper health-seeking behaviour is essential for better health outcomes. Inappropriate health-seeking behaviour can influence worse health outcomes to increase morbidity and mortality [5]. There are significant barriers to seeking eye care services for children. These include lack of availability of eye care services, lack of awareness about the need and availability of services, financial constraints, utilization of time to receive such services and improper referral by general practitioners and paediatricians [6–8]. A belief in traditional medicine and taboos have also been barriers to seeking eye care services [6,9]. Knowing eye disease, people neglect to seek eye care services, and there exists a large gap between mass awareness and treatment practices for common eye diseases in low and middle-income countries [10,11].

Childhood blindness and visual impairments constitute a significant proportion of ocular morbidities in developing countries, which can be treated and prevented with strategic efforts [12]. The prevention of ocular morbidities among children requires early diagnosis and treatment. Two different studies conducted in Chennai, India and Nigeria indicated that seeking eye care for children depends on parental awareness and perception of their children's eye health [13,14].

Additionally, demographic factors affect the utilization of eye care services, such as age, race, gender, socioeconomic status, education, knowledge about eye diseases, and services available for eye care [15].

Like in other low and middle-income countries, ocular morbidity is an established public health problem in Bangladesh. Relevant recent epidemiological data on the burden of this existing problem is scarce. A study showed that the prevalence of childhood blindness was 6.3 per 10,000 among children aged $\leq$ 15 years [1]. Limited population-based studies consider both the blind and non-blind causes of childhood visual problems. Moreover, almost no data have been found on the health-seeking behavioural pattern of the caregivers/ parents towards childhood ocular problems. This study was designed to explore the health-seeking behaviours of parents in Bangladesh whose children have ocular issues. The findings of this study facilitated the development of a national strategy for attaining better health-seeking behaviours of parents to reduce the consequences of childhood visual problems.

## Materials and methods

It was a cross-sectional, mixed-method study. Both qualitative and quantitative methods were used to explore the eye health care-seeking practices of caregivers/parents having children with ocular morbidity. A quantitative face-to-face interview was carried out among the caregivers/parents having children with ocular morbidity and a qualitative interview was carried out with the service care providers and local leaders in three unions (the lowest administrative geographical area comprised 30,000–50,000 population) of the Raiganj Upazila, under the

Sirajganj District of Bangladesh. The study period was from January to April 2017. The details of the study methods, including study population and data collection procedures, have been described elsewhere [1].

## Patient and Public Involvement

Both the parents/ caregivers of the children with ocular morbidities and disabilities and service providers were interviewed in this study. Initially, a quantitative face-to-face interview was conducted by trained field workers (3 field supervisors and 15 data collectors) to identify the children with visual problems. The data collectors identified five hundred and seventy cases among 39,351 children at the field level. All cases were invited to join another session organized by the local office of CIPRB. A team of 5 paediatric ophthalmologists then confirmed the screened-out cases. The ophthalmologist kept a record of the principal and contributing reasons for the children's visual impairments and blindness. In total, 198 cases were confirmed by the team as children having ocular morbidity.

In the next stage, parents or caregivers of 198 patients were interviewed to explore their health-seeking behavior for children with ocular illness. Face-to-face interviews were conducted using a semi-structured questionnaire. A team of 2 trained data collectors conducted the interviews. In total, 198 caregivers participated in the study. Socio-demographic information of the respondents was collected from the database of the CIPRB surveillance system. All individuals in the surveillance area have their unique identification numbers. After collecting specific information, it was merged with the required socio-demographic variables of the existing population database.

A total of six Key Informant Interviews (KIIs) were conducted using a topic guide containing specific questions about the barriers of eye health-seeking behaviours and recommendations to mitigate the problem with the service care providers and local leaders. The 6 KIIs were conducted as we reached the thematic saturation after finishing six interviews. Service providers were from tertiary hospitals and primary care centres; local leaders were from the above mentioned Upazila and Union. These interviews aimed to gather insights and perspectives from service providers involved in caring for paediatric patients and local leaders involved in the community, where we took the quantitative data. The selection of service providers and local leaders as key informants was based on their expertise and experience in providing eye care services to paediatric patients and the community representative who best knew the condition of the parents or caregivers of the above-mentioned Upazila and Union. By including professionals from both tertiary hospitals and primary care centres, as well as local leaders, a diverse range of perspectives and insights were obtained, contributing to a more comprehensive understanding of the topic under investigation. The qualitative interviews were conducted by the researcher and a research assistant using the online platform, as the interviewees were not available for the face-to-face interview. Most interviewees agreed to give their interview in the evening due to their busy schedules. Each interview lasts 30 to 40 minutes. The whole interview process took five days to complete in March 2017. Most of the interviewees were very cooperative throughout the interview and they chose a close room from their end so that they could maintain an environment where no disturbance occurred during the whole process.

During the interviews, all participants were assured of confidentiality and anonymity. The interviews were audio-recorded with the consent of the participants, and detailed hand notes were taken to capture additional information and nuances where appropriate. This combination of audio recording and note-taking allowed for comprehensive documentation of the interviews and ensured accuracy in capturing the participants' responses.

## Data analysis

A descriptive analysis was carried out to describe the population characteristics both for the children and the households. In this analysis, we considered the age and gender of the children, age, literacy, sources of income earning of the household head, and economic status of the head of households' independent variables. The type of service provider was considered as dependent variable. Results were shown as percentages according to the type of service provider. A Chi-square test was done to see the statistical significance between dependent and independent variables. $P < 0.05$ was considered statistically significant. Seeking service and seeking service from qualified service providers were the dependent variables. Age and sex of the children, age, education, sources of income earning of the household head, and household economic condition were considered as independent variables. In the qualitative part, all audio recordings were transcribed, and from them, the result was zest out. Thematic analysis was carried out for the qualitative part.

## Analysis of qualitative data

The discussions were transcribed and then coded independently by the two research assistants carefully and accurately. Through content analysis, major themes were identified, coded, and categorized [16]. Validity was ensured by comparing the researchers' findings among themselves and with those of an independent investigator. No major inter-rater inconsistencies were found. After sorting and categorizing the responses, excerpts from the transcripts were chosen to illustrate the summary statements, which were also used to validate the findings. The transcription was made in Bangla and then translated into English. We took samples from other researcher/s and then performed translation and back-translation to ensure the quality. The socio-demographic information of the interviewee is listed down below:

| Participants ID | Sex | Age | Education |
|---|---|---|---|
| KII 01 | Male | 55 | Medical degree |
| KII 02 | Male | 47 | Medical degree |
| KII03 | Male | 40 | Medical degree |
| KII 04 | Female | 40 | Medical degree |
| KII 05 | Male | 60 | Bachelor's degree |
| KII 06 | Male | 60 | Bachelor's degree |

## Operational definitions

**Ocular morbidities.** This encompassed childhood blinding and non-blinding conditions. The definition used for childhood blindness is described elsewhere [1]

**Qualified service providers.** Public and private hospitals, clinics and health facilities, and private registered physicians.

**Non-qualified service providers.** These included government and NGO's field-level health workers, village doctors, drug sellers, herbal/homoeopathic practitioners, traditional/religious healers and others.

**Wealth index.** The definition of the wealth index used for this study has already been published elsewhere [17].

**Literacy of household head.** A household head who attended no school education, informal education and only one year of schooling was considered 'No Education'. A primary level of education was defined as when the household head had five completed years of schooling.

Those who attended more than five years of school education were designated as 'SSC and above'.

**Occupation of household head.** The unskilled worker group included those involved in house duties, unskilled labour, servants and retired, whereas the skilled worker group included the categories of service holder and skilled labourer.

**Sources of income earning of the household head.** Skilled worker refers to a job where s/he requires judgement to perform the assigned duties, including service sectors. The unskilled worker refers to a job requiring unimportant or no judgement to perform the assigned duties, including household work, and untrained work without any skills.

## Ethical consideration

Ethical clearance for this study was received from the Ethical Review Committee of the Centre for Injury Prevention and Research, Bangladesh (CIPRB/ERC/2016/006). Written consent of all caregivers was gained before conducting the study.

## Results

In this study, the proportion of boy children (51.4%) is quite similar to that of girl children (48.6%). The majority of the children (70.0%) were between 5 and 15 years of age. About two-thirds of the children's household heads were in the young age group (64.7%). Most of the household heads were illiterate (61.6%) and belonged to poor socioeconomic conditions (50.0%), as shown in (Table 1).

**Table 1. Population characteristics (N = 39351).**

| Variables | Frequency | Percentage |
|---|---|---|
| **Children's characteristics** | | |
| **Sex of the child** | | |
| Girl | 19,140 | 48.6 |
| Boy | 20,211 | 51.4 |
| **Age of the child (in years)** | | |
| 5 and Above | 27,545 | 70.0 |
| Under-5 | 11,806 | 30.0 |
| **Literacy of household head** | | |
| No education | 24,243 | 61.6 |
| Primary | 11,856 | 30.1 |
| SSC and above | 3252 | 8.3 |
| **Age of household head** | | |
| <35yrs | 25,462 | 64.7 |
| 36–59 | 10,968 | 27.9 |
| 60 and above | 2921 | 7.4 |
| **Sources of income earning of the household head** | | |
| Unskilled work | 15,005 | 38.1 |
| Agriculture | 10,254 | 26.1 |
| Business | 4929 | 12.5 |
| Skilled work | 9163 | 23.3 |
| **Wealth index** | | |
| Poor | 19,678 | 50.0 |
| Middle | 9843 | 25.0 |
| Rich | 9830 | 25.0 |

**Barriers to the Uptake of Eye Health Services in a rural community of Bangladesh: A community based cross-sectional survey**

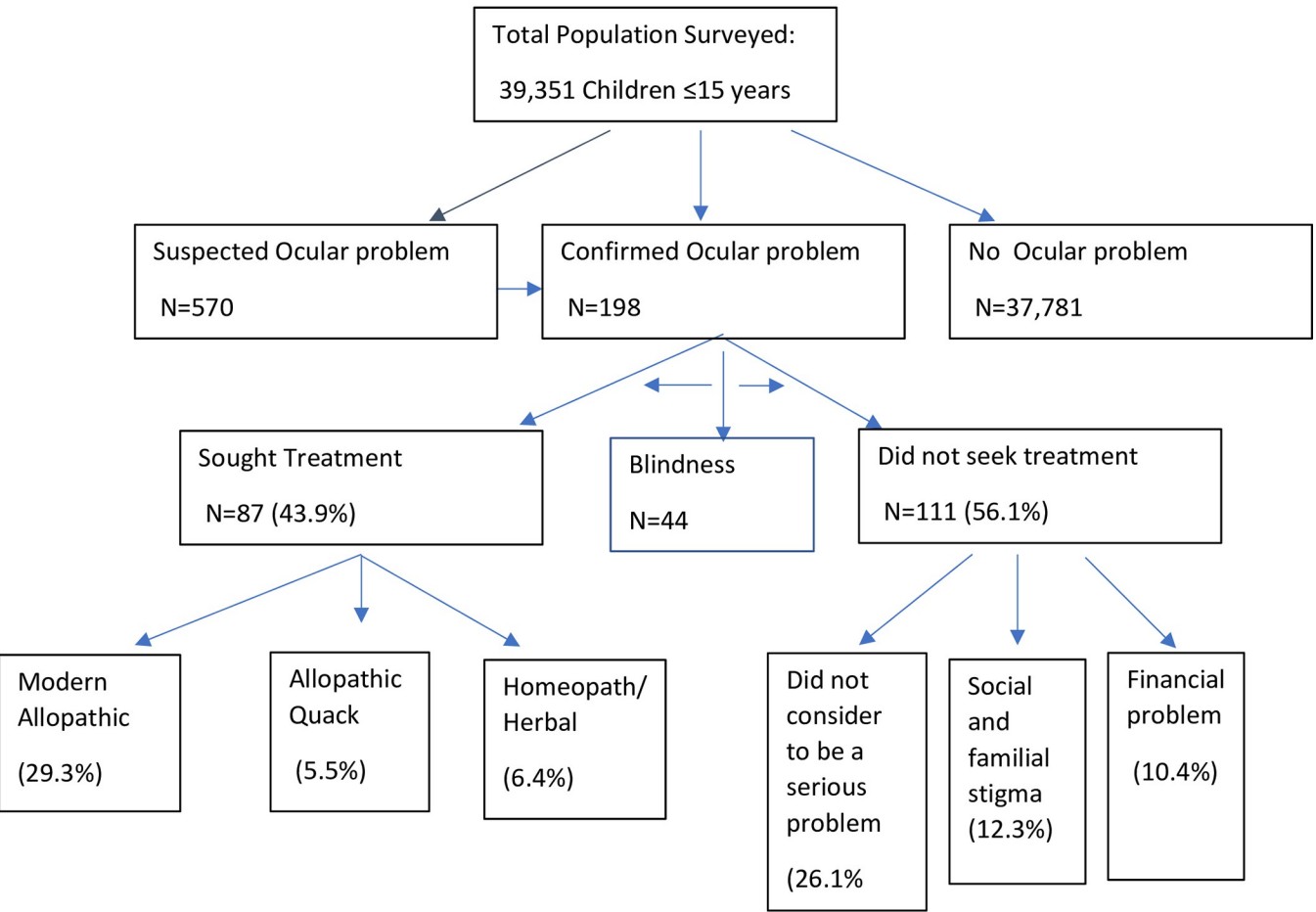

**Fig 1. Health-seeking pattern for childhood ocular morbidities in a rural community of Bangladesh.**

A total of 198 confirmed cases of childhood ocular morbidities, including blindness, were identified; among them, only 87 (43.9%) sought healthcare for their ocular morbidities—those who sought health care more than one-third sought non-qualified healthcare providers. Among the 43.9%, around 14% of providers were herbal/homoeopathic practitioners and about 5% were medicine shopkeepers (Figs 1 and 2).

Health-seeking behaviour was found to be better among wealthier families. A higher proportion of families in the more affluent group sought health care for their children with ocular problems. It was 55.3% and 36% among wealthy and low-income families, respectively. A higher proportion of wealthy families sought care from qualified service providers than low-income families. It was 90.5% and 52.5%, respectively. The difference was statistically significant in both of the cases. Similarly, a higher proportion of families sought care for their children when a household is literate than a family with an illiterate household head.

Educated household heads chose qualified service providers for their children at a higher rate than illiterate household heads. Seeking service and choosing a qualified service provider found when a child was male in gender (Table 2).

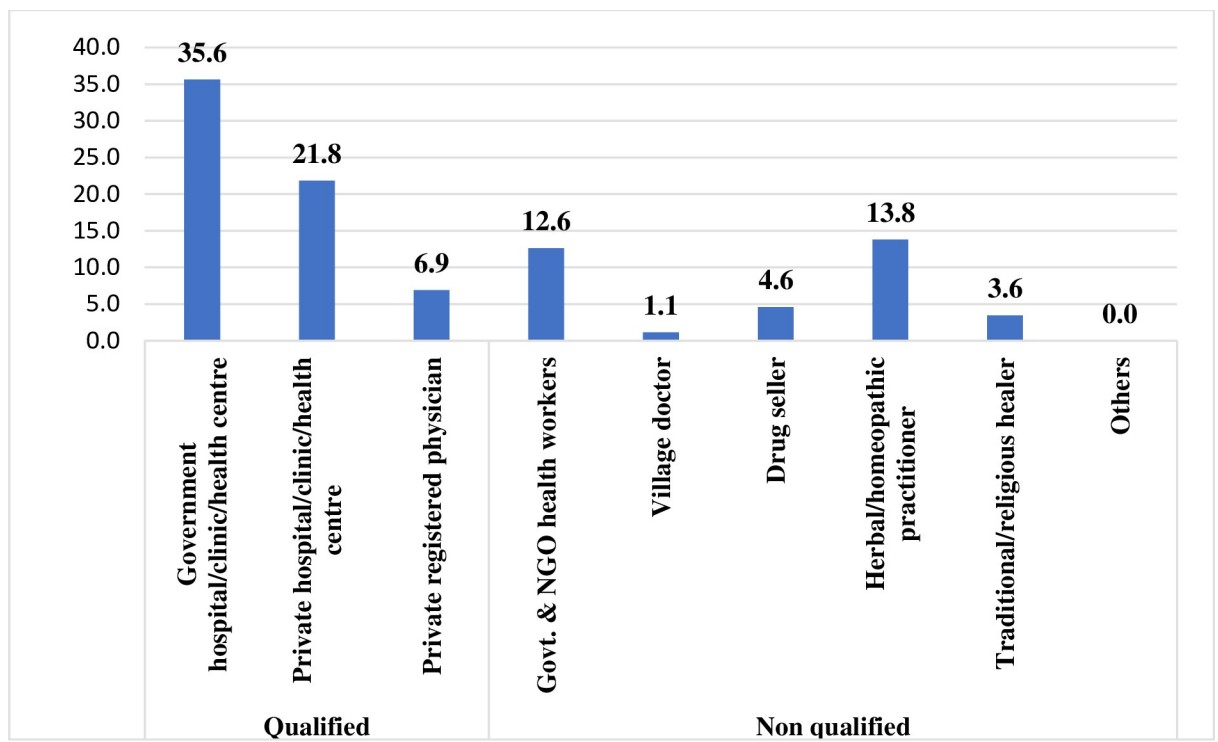

**Fig 2. Health-seeking pattern for childhood eye problems in rural Bangladesh.**

The leading barriers responsible for not seeking services were lack of knowledge, social stigma and financial constraints, with proportions of 51.4%, 22.5% and 18.9%, respectively as shown in Fig 3.

## Qualitative findings

During the thematic analysis, three main themes emerged from the data. Theme one highlighted the lack of knowledge among parents regarding eye health care seeking for their children. Themes two and three focused on the lack of awareness and economic constraints, which were also reflected in the quantitative results.

**Lack of knowledge about the disease and its consequence.**   All the participants in the qualitative interviews stated that lack of knowledge was the key barrier to seeking eye health care. They mentioned that parents often have insufficient knowledge about eye diseases in children, leading to a lack of awareness and delayed care-seeking. Some parents were unaware of where to seek paediatric eye care. One of the service provider participants stated.

"Some parents are aware that their children have some eye problem, but they do not know where to seek paediatric eye care". (PO 3)

Another participant highlighted the impact of parents' educational background, stating, "Educational background of the parents is another main barrier to seek eye health care. In many cases, physicians referred them to ophthalmologists to seek eye care but due to their education, they did not give it any importance and delayed to seek eye care for their children". (PO 5)

**Lack of awareness about children's ocular illness.**   Lack of awareness about children's ocular illness was another barrier to seeking eye care. Participants narrated that most of the time, the parents were ignorant about their children's eyes due to proper awareness. The

**Table 2. Socio-demographic characteristics and care seeking pattern of childhood eye problem.**

| | Service received | | | Type of Service received | | |
|---|---|---|---|---|---|---|
| | Yes | No | P | Non-qualified | Qualified | P |
| **Age of child** | | | | | | |
| 5 & Above | 43.7 | 56.3 | 0.90 | 32.3 | 67.7 | 0.30 |
| Under-5 | 44.6 | 55.4 | | 44.0 | 56.0 | |
| **Gender of child** | | | | | | |
| Girl | 38.5 | 61.5 | 0.09 | 37.1 | 62.9 | 0.49 |
| Boy | 48.6 | 51.4 | | 34.6 | 65.4 | |
| **Education of HH head** | | | | | | |
| No education | 41.4 | 58.6 | 0.14 | 41.5 | 58.5 | 0.35 |
| Primary | 37.5 | 62.5 | | 25.0 | 75.0 | |
| SSC and above | 57.9 | 42.1 | | 27.3 | 72.7 | |
| **Sources of income earning of the household head** | | | | | | |
| Unskilled work | 47.8 | 52.2 | 0.80 | 31.3 | 68.7 | 0.31 |
| Agriculture | 40.3 | 59.7 | | 44.0 | 56.0 | |
| Business | 40.0 | 60.0 | | 16.7 | 83.3 | |
| Skilled work | 46.2 | 53.8 | | 44.4 | 55.6 | |
| **Age of HH head** | | | | | | |
| 17–35 yrs | 38.9 | 61.1 | 0.13 | 41.2 | 58.8 | 0.13 |
| 36–59 yrs | 52.1 | 47.9 | | 36.0 | 64.0 | |
| 60yrs & above | 57.9 | 42.1 | | 9.1 | 90.9 | |
| **Economic condition** | | | | | | |
| Poor | 36.0 | 64.0 | 0.04 | 47.5 | 52.5 | 0.01 |
| Middle | 53.1 | 46.9 | | 38.5 | 61.5 | |
| Rich | 55.3 | 44.7 | | 9.5 | 90.5 | |

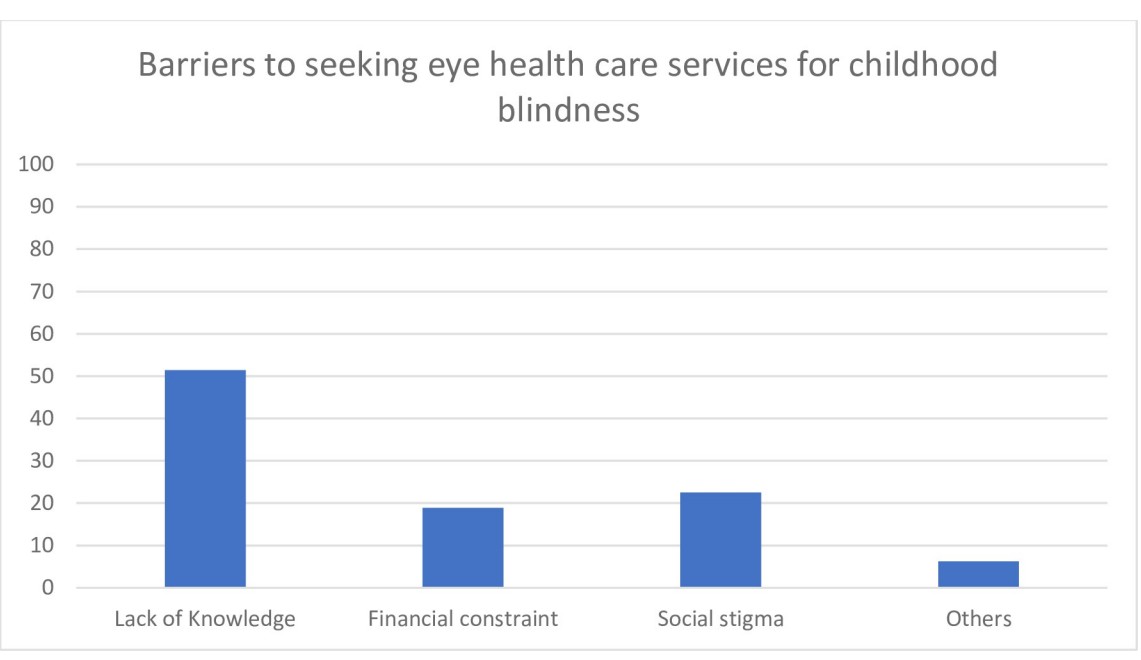

**Fig 3. Barriers to seeking health care services for childhood blindness in a rural community in Bangladesh. (N = 111).**

parents thought that it would be okay when the children were adults. One of the participants also expressed that.

"Child cannot complain and mothers cannot understand the problem as they were not well aware of paediatric eye problem". (PO 1)

**Economic constraints.** Qualitative findings revealed that economic constraints were a significant barrier to health seeking behaviour. Respondents highlighted that the high out-of-pocket expenditure prevented rural people from seeking care for their children's eye problems.

One respondent explained, "As our out-pocket expenditure was high, people from rural areas did not seek paediatric eye care as they had to pay a lot of money for transportation as well as for treatment purposes". (PO 2)

Another participant emphasized the lack of a strong referral system, leading parents to incur high costs when they finally sought treatment, often requiring them to take out loans to continue the treatment: "As we do not have a strong referral system, parents go here and there for the treatment purposes, and ultimately, when they go to the right place for treatment, their cost exceeds their budget and they have to take out a loan to continue the treatment". (PO 5)

These qualitative findings highlight the critical barriers of lack of knowledge, lack of awareness, and economic constraints that hinder parents' ability to seek appropriate eye health care for their children. Addressing these barriers is crucial for improving pediatric eye care utilization and ensuring timely interventions for children's ocular health.

## Discussion

This study was designed to explore the health-seeking behavior of parents with children with ocular morbidities and disabilities. This study was conducted among parents who had a child with ocular morbidity and service providers from primary and tertiary eye care. We explored the health-seeking behavior of the parents of those 198 children with ocular problems and the service providers who are giving eye care at the primary and tertiary care levels.

In this study, more than half of the parents did not seek health care for their children with ocular morbidities. A similar finding was observed in a study conducted in Nigeria [18]. A national survey found that parents sought health care at a lower rate for common illnesses like the common cold and diarrhea [19]. However, better health-seeking behaviour was found in a study conducted in an urban slum in Dhaka [20]. In our study, an association was found between the health-seeking behavior of the parents and their economic condition and educational qualifications. Parents seek health care for their children at a higher rate when their economic condition improves. The wealthy family also chose qualified health care providers for their children. In Bangladesh, a similar association was observed between health-seeking behavior for their children and other health-related events [21]. A study in India shows that lack of money is one of the significant barriers to seeking eye care for their children [22]. Also in this study, economic constraints are the major barrier to seeking eye care, as found in the qualitative part. A study conducted in Nigeria supported the same result [23]. Educated parents who seek health care for their children at a higher rate also choose qualified health care providers to care for their children. In the qualitative exploration of this study, service providers mentioned that a lack of knowledge about the disease is a major barrier to seeking health care. A similar finding was observed in the health-seeking behavior of parents for childhood burn injuries in Bangladesh [21]. The finding was consistent with other studies in similar settings [20,24,25]. Compared to parents with younger children, parents who have children aged five years or more sought health care at a higher rate. Children of a higher age can complain about their problem, or symptoms become more severe with the progression of age. On the other hand, children in lower age groups cannot complain, and their issues remain

unaddressed. However, early detection and treatment are essential for a better outcome of an illness. Some interventions can be taken to make parents aware of early-age symptoms. We found boys are more likely to get health care and care from quality healthcare providers than girls. However, the finding was not statistically significant. A study conducted in Africa found that the care-seeking pattern is strongly related to the perceived severity of the illness [9].

There are many possible psychosocial and economic barriers to seeking eye care services for children. For example, studies in India and Africa have shown that financial constraints are a major barrier for parents seeking eye care services for their children [9,22]. In this study, caregivers' lack of knowledge, lack of awareness, financial constraints, and social taboo about childhood ocular problems were deemed the major barriers to seeking eye health services in Bangladesh. Studies conducted in other countries about parents' awareness and perceptions of childhood ocular problems have also illustrated that parents' knowledge and understanding of the aetiology of eye health problems can be lacking [14]. Such factors may influence caregivers' healthcare behaviours and militate against seeking help for their children [14].

More than half of the parents did not seek eye care services for their children. Of those who sought health care, one-third sought care from an unqualified service provider. Socio-demographic factors, including lack of awareness, play an important role in seeking eye care services. A comprehensive strategy needs to be developed to address childhood ocular morbidities, including childhood blindness. Two significant factors must be addressed. There need to be comprehensive screening facilities available at the community level to facilitate the early detection of eye health problems. Additionally, caregivers must become more aware of possible eye health problems and the merits of early interventions when problems occur.

Bangladesh has successfully integrated primary eye health care for children into the country's Integrated Management of Childhood Illness (IMCI) programme in 2018, where identifying eye problems and strong referral mechanisms to the eye department at district hospital were developed. Community health workers attached to facilities delivering IMCI were also engaged to promote awareness about eye conditions in children in the community.

The findings of this study will contribute to the formulation of effective policies aimed at enhancing health-seeking behaviour, particularly concerning the ocular health of children in Bangladesh.

## Strength and limitation

This was so far the first community-based study conducted in Bangladesh to find out the health-seeking behaviour of caregivers and parents of children who have ocular problems. Our findings have a significant impact on preventing childhood ocular problems. Policymakers can utilize this knowledge to formulate an action plan to reduce childhood blindness by integrating intra- and inter-sectoral interventions. Though this is the first community-based study and this study was conducted in only one sub-district, it is not a nationally representative study.

## Conclusion

More than half of the caregivers did not seek any eye care services for their children. Socio-demographic factors, including a lack of awareness, play an essential role in the health-seeking behaviour of the parents. Improved healthcare-seeking behaviours may be facilitated by improving caregivers' knowledge and attitudes towards eye health problems and how problems may be addressed. There will be increased opportunities to reduce childhood ocular morbidity and disability and prevent unnecessary visual problems and blindness in low resource settings in developing countries.

## Supporting information

**S1 Checklist. STROBE statement—barriers to the uptake of eye health services in a rural community of Bangladesh: A community based cross-sectional survey.**
(DOC)

**S1 File. Original data set of this study.**
(SAV)

**S2 File. Original transcribe file of this study (Bangla Version).**
(DOCX)

**S3 File. Original translation file of this study (English version).**
(DOCX)

## Acknowledgments

This study was conducted in the surveillance area of the Centre for Injury Prevention and Research Bangladesh (CIPRB). The CIPRB provided administrative and technical support in the data collection and data management procedures. We are also grateful to the ophthalmologists who took part in the visits for the detection of the confirmed cases. We are thankful to the members of the office of the Director General of Health Services for their technical support in the validation of our research materials.

## Author Contributions

**Conceptualization:** A. H. M. Enayet Hussain, Labida Islam, Saidur Rahman Mashreky, Koustuv Dalal.

**Data curation:** A. H. M. Enayet Hussain.

**Formal analysis:** A. H. M. Enayet Hussain, Labida Islam, Saidur Rahman Mashreky, Koustuv Dalal.

**Methodology:** A. H. M. Enayet Hussain, Labida Islam.

**Supervision:** Koustuv Dalal.

**Writing – original draft:** A. H. M. Enayet Hussain, Labida Islam.

**Writing – review & editing:** Saidur Rahman Mashreky, A. K. M. Fazlur Rahman, Eija Viitasara, Koustuv Dalal.

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
