## [Decision Letter · Decision Letter 0]

16 Aug 2023

PONE-D-23-17474Barriers to the Uptake of Eye Health Services of the Children in rural Bangladesh: A community based cross-sectional surveyPLOS ONE

Dear Dr. Hussain,

Thank you for submitting your manuscript to PLOS ONE. After careful consideration, we feel that it has merit but does not fully meet PLOS ONE’s publication criteria as it currently stands. Therefore, we invite you to submit a revised version of the manuscript that addresses the points raised during the review process.

ACADEMIC EDITOR:

Dear authors,

The paper addressed important issues. However, the paper needs to be strengthened in communication and scientific explanations in methodology and reporting. Hence, there is a need for major revision for reassessment. 

With Ranjit

We look forward to receiving your revised manuscript.

Kind regards,

Ranjit Kumar Dehury

Academic Editor

PLOS ONE

Additional Editor Comments:

Dear authors,

The paper addressed important issues. However, the paper needs to be strengthened in communication and scientific explanations in methodology and reporting. Hence, there is a need for major revision for reassessment.

With Ranjit

Reviewers' comments:

Reviewer's Responses to Questions

**Comments to the Author**

1. Is the manuscript technically sound, and do the data support the conclusions?

Reviewer #1: Yes

Reviewer #2: Partly

2. Has the statistical analysis been performed appropriately and rigorously? 

Reviewer #1: Yes

Reviewer #2: No

3. Have the authors made all data underlying the findings in their manuscript fully available?

Reviewer #1: Yes

Reviewer #2: Yes

4. Is the manuscript presented in an intelligible fashion and written in standard English?

Reviewer #1: Yes

Reviewer #2: Yes

5. Review Comments to the Author

Reviewer #1: Dear Author

This is a very important manuscript with notable samples and originality.

The title is concise, no question.

The introduction provides the necessary background and explains why it is essential to do the study.

The METHODS and MATERIALS descriptions are correct. The authors state having IRB approval.

The Results section is consistent with the Methods and Materials section and adequately reported with numbers and distribution values.

DISCUSSION begins with a brief outline of the principal findings of the study. The findings were put into context and provided adequate comparisons with previous studies. A brief mention of clinical relevance sums it up.

FIGURE is able to be read as a standalone element of the manuscript. The table is clear and simple to comprehend.

Reference checking could not be completed due to a system error. Please check and/or verify again.

I have a few queries and/or comments. Please make the necessary corrections.

Congratulations to the authors!

Al

MD. Al-Amin Bhuiyan, PhD MPH BDS

Reviewer #2: How did the author categorize the respondent’s occupation? Skilled and unskilled is not a part of occupation. Author can consider them as wage workers.

explain about non-qualified and qualified services. how can a reader differentiate them?

The qualitative findings do not follow the proposed analysis methods. Author can quote the field narratives for depth understanding of the issues.

what are the government intervention to mitigate the issue. What is the state mechanism promoting awareness and knowledge about children's ocular illness.

Author should discuss their policy and programme for finding the gaps.

6. PLOS authors have the option to publish the peer review history of their article (what does this mean?). If published, this will include your full peer review and any attached files.

Reviewer #1: **Yes: **Md Al Amin Bhuiyan

Reviewer #2: No

---

## [Author Response · Author response to Decision Letter 0]

25 Sep 2023

Academic Editor: Thank you for your positive feedback. As per your suggestion Data file uploaded as a supporting information. 

1.Reviewer 1

Reference checking could not be completed due to a system error. Please check and/or verify again. 

-Thank you for your valuable feedback. We have revised the reference issues in the manuscript and the changes are in the following line number 44, 358 and 366.

I have a few queries and/or comments. Please make the necessary corrections.

-Thank you once again. Corrections are made in the following document below as well as in the manuscript with track change. 

2.Reviewer 2

How did the author categorize the respondent’s occupation? Skilled and unskilled is not a part of occupation. Author can consider them as wage workers. 

-Thank you for your comment in the manuscripts. Revision was made in the manuscripts in both table 1 and 2. Changes made in the line number 176 and 200. And an operational definition was described in the operational definition section as “Skilled worker refers to a job where s/he require judgement to perform the assigned duties including service sectors. The unskilled worker refers to job requiring unimportant or no judgement to perform the assigned duties including household works, untrained work without any skills” in the line number 161 to 164. 

Explain about non-qualified and qualified services. how can a reader differentiate them? 

-Thank you for your response. As per the operational definition of the manuscripts; Public and private hospitals, clinics and health facilities, and private registered physicians are defined as qualified service providers in this paper. And government and NGO's field-level health workers, village doctors, drug sellers, herbal/homoeopathic practitioners, traditional/ religious healers and others are defined as non- qualified service providers. Operational definition of this two was described in the 146 and 148 no line in the manuscript.

The qualitative findings do not follow the proposed analysis methods. Author can quote the field narratives for depth understanding of the issues. 

-Thank you for this comment. Revision was made according to your comments in the line number 134 to 142 and 211 to 249.

what are the government intervention to mitigate the issue. What is the state mechanism promoting awareness and knowledge about children's ocular illness. Author should discuss their policy and programme for finding the gaps.

-Bangladesh has successfully integrated primary eye health care for children into the country's Integrated Management of Childhood Illness (IMCI) programme in 2018, where identifying eye problems and strong referral mechanisms to the eye department at district hospital were developed. Community health workers attached to facilities delivering IMCI were also engaged to promote awareness about eye conditions in children in the community. The findings of this study will contribute to the formulation of effective policies aimed at enhancing health-seeking behaviour, particularly concerning the ocular health of children in Bangladesh. This section was discussed in the discussion section in the line number 308 to 313.

Other comments in the pdf file by Authors

-Thank you for all the comments in our manuscript. Necessary changes are listed below according to your comments.

Title: Revision was made according to your suggestion. Correction was done in the line number 2 & 3. Title: Barriers to the uptake of eye health services of the children in rural Bangladesh: A community based cross-sectional survey.

Abstract: Necessary changes were made in the line number 19 and 32 to 35.

Line number 5: KoustuvDalal: Space was given in between the name in the line number 5.

References: Uniformity of the references were made in the document.

56 no line: Space was given before A in the manuscripts in the line number 63.

88 no line: Revision was made according to your suggestion. Caregiver was used in the whole manuscripts in the line number of 90, 96, 106 and 320.

Face-to-face interviews were conducted using a structured questionnaire: Semi-structured questionnaire was used in the manuscript according to your comments in the line number 108.

Analysis of qualitative data: Revision was made in the manuscripts in the line number 134 to 142 and 211 to 249.

Table 1 and table 2: In table 1 and 2 sex, age and occupation were revised accordingly. Table 1 and 2 were in the line number 176 and 200.

198 no line: Small letter k was used in the line number 216.

199 no line: Rephrasing was done in the line number 217 to 218.

208 no line: Lack of awareness was used in the line number 232.

Discussion: Discussion was thoroughly checked for grammatical error and revised in the line number 251 to 316. 

Strength and limitation: Strength and limitation were revised according to your comments in the line number 318 to 325.

283 no line: In the conclusion a was given before lack of awareness in the line number 328.

Figure 3: Title was given and axis was changes according to your suggestion.

---

## [Decision Letter · Decision Letter 1]

10 Oct 2023

PONE-D-23-17474R1Barriers to the uptake of eye health services of the children in rural Bangladesh: A community based cross-sectional surveyPLOS ONE

Dear Dr. Hussain,

Thank you for submitting your manuscript to PLOS ONE. After careful consideration, we feel that it has merit but does not fully meet PLOS ONE’s publication criteria as it currently stands. Therefore, we invite you to submit a revised version of the manuscript that addresses the points raised during the review process.

ACADEMIC EDITOR:

Dear authors,

After assessment it is found that still some components have to be improved to bring publishable quality. Hence, you have to revise according to the comments of the reviewers.

With regards,

Ranjit

We look forward to receiving your revised manuscript.

Kind regards,

Ranjit Kumar Dehury

Academic Editor

PLOS ONE

Journal Requirements:

Additional Editor Comments:

Dear authors,

After assessment it is found that still some components have to be improved to bring publishable quality. Hence, you have to revise according to the comments of the reviewers.

With regards,

Ranjit

Reviewers' comments:

Reviewer's Responses to Questions

**Comments to the Author**

1. If the authors have adequately addressed your comments raised in a previous round of review and you feel that this manuscript is now acceptable for publication, you may indicate that here to bypass the “Comments to the Author” section, enter your conflict of interest statement in the “Confidential to Editor” section, and submit your "Accept" recommendation.

Reviewer #1: All comments have been addressed

Reviewer #2: (No Response)

2. Is the manuscript technically sound, and do the data support the conclusions?

Reviewer #1: Yes

Reviewer #2: Partly

3. Has the statistical analysis been performed appropriately and rigorously? 

Reviewer #1: Yes

Reviewer #2: Yes

4. Have the authors made all data underlying the findings in their manuscript fully available?

Reviewer #1: Yes

Reviewer #2: Yes

5. Is the manuscript presented in an intelligible fashion and written in standard English?

Reviewer #1: Yes

Reviewer #2: Yes

6. Review Comments to the Author

Reviewer #1: (No Response)

Reviewer #2: The author needs to work on qualitative parts. The field narrative is still missing in the corrected version. The result section should be follow the proposed methodology.

7. PLOS authors have the option to publish the peer review history of their article (what does this mean?). If published, this will include your full peer review and any attached files.

Reviewer #1: No

Reviewer #2: No

---

## [Author Response · Author response to Decision Letter 1]

14 Nov 2023

Academic Editor 

-Thank you for your positive feedback. Reference lists are revised according to your recommendation and retracted paper has removed. Changes are made in the line number 379 to 451

Reviewer 

2. The author needs to work on qualitative parts. The field narrative is still missing in the corrected version. The result section should be follow the proposed methodology. 

-Thank you for your valuable feedback. We have revised the whole qualitative parts and the changes made in the following line number 87 to 89, 112 to 139, 162 to 165 and 231 to 273.

---

## [Editor Report · Decision Letter 2]

21 Nov 2023

Barriers to the uptake of eye health services of the children in rural Bangladesh: A community based cross-sectional survey

PONE-D-23-17474R2

Dear Dr. Hussain,

We’re pleased to inform you that your manuscript has been judged scientifically suitable for publication and will be formally accepted for publication once it meets all outstanding technical requirements.

Kind regards,

Ranjit Kumar Dehury

Academic Editor

PLOS ONE

Additional Editor Comments (optional):

Dear authors,

The manuscript is now of publishable quality. Hence, it is accepted for publication.

With regards,

Ranjit
---

## [Editor Report · Acceptance letter]

28 Nov 2023

PONE-D-23-17474R2 

Barriers to the uptake of eye health services of the children in rural Bangladesh: A community-based cross-sectional survey 

Dear Dr. Hussain:

I'm pleased to inform you that your manuscript has been deemed suitable for publication in PLOS ONE. Congratulations! Your manuscript is now with our production department. 

Kind regards, 

on behalf of

Dr. Ranjit Kumar Dehury 

Academic Editor

PLOS ONE